# A Multi-Task Corpus for Assessing Discourse Coherence in Chinese Essays: Structure, Theme, and Logic Analysis

**Hongyi Wu[1], Xinshu Shen[1], Man Lan[1,2,*], Yuanbin Wu[1], Xiaopeng Bai[3], Shaoguang Mao[4]**

[1]School of Computer Science and Technology, East China Normal University, Shanghai, China
[2]Shanghai Institute of AI for Education, East China Normal University, Shanghai, China
[3]Department of Chinese Language and Literature, East China Normal University, Shanghai, China
[4]Microsoft Research Asia, Beijing, China
{hongyiwu,xinshushen}@stu.ecnu.edu.cn, {mlan,ybwu}@cs.ecnu.edu.cn
xpbai@zhwx.ecnu.edu.cn, shaoguang.mao@microsoft.com

## Abstract

This paper introduces the **C**hinese **E**ssay **D**iscourse **C**oherence **C**orpus (**CEDCC**), a multi-task dataset for assessing discourse coherence. Existing research tends to focus on isolated dimensions of discourse coherence, a gap which the CEDCC addresses by integrating coherence grading, topical continuity, and discourse relations. This approach, alongside detailed annotations, captures the subtleties of real-world texts and stimulates progress in Chinese discourse coherence analysis. Our contributions include the development of the CEDCC, the establishment of baselines for further research, and the demonstration of the impact of coherence on discourse relation recognition and automated essay scoring. The dataset and related codes is available at https://github.com/cubenlp/CEDCC_corpus.

## 1 Introduction

Discourse coherence, a fundamental aspect of both language comprehension and generation, involves understanding how words, phrases, sentences, and paragraphs within a text are interconnected to create a cohesive and coherent message. Due to its significance, it has been widely incorporated into various natural language processing (NLP) tasks, such as automated essay scoring (Farag et al., 2018), machine translation (Xiong et al., 2019) and text generation (Guan et al., 2021).

Existing discourse coherence analyses present various frameworks for understanding coherence realization, most of which can be grouped into three categories: (a) **Overall Coherence Grading**, which assigns a coherence score to a given text, based on its overall logical flow and consistency; (b) **Topical Continuity Modelling**, which focuses on analysing the distribution and relationship of topics within discourses; and (c) **Discourse Relationship Recognition**, which involves detecting semantic relations between two segments of a text.

Each of these elements highlights a distinct aspect of discourse coherence. Specifically, for coherence grading, Lai and Tetreault (2018) assert that inherent coherence is a fundamental feature of human-written texts, crucial for the global comprehension of an essay and its quality assessment. In terms of topical continuity, Amoualian et al. (2017) argue that a coherent text should maintain a logical progression of ideas, eschewing abrupt topic transitions. They primarily utilize topic models to extract and designate topics within a text, thereby quantifying topical coherence. Furthermore, Feng et al. (2014) suggest that text coherence is closely tied to its discourse structure and relations.

Despite the multifaceted nature of discourse coherence, current research often examines isolated dimensions, overlooking their interplay. For example, most coherence grading studies (Shen et al., 2021; Flansmose Mikkelsen et al., 2022) often perceive coherence as an automatic scoring task, neglecting aspects like topical continuity and discourse relations, which impairs the interpretability of results. Although some studies aim to integrate these dimensions, progress has been hindered by the lack of a comprehensive dataset. For instance, methods proposed by Shrivastava et al. (2018) and Lin et al. (2011) operate in unsupervised or artificially structured contexts, but these fail to capture the complexity of real-world texts. Furthermore, the scarcity and limited diversity of Chinese discourse coherence datasets have somewhat constrained advancements in this field.

To address the shortcomings of existing research, we introduce the **C**hinese **E**ssay **D**iscourse **C**oherence Corpus (**CEDCC**), a multi-task dataset designed to assess essay discourse coherence. The CEDCC, sourced from various middle schools, encompasses a diverse range of topics, genres, and regions. Each essay has been meticulously annotated by linguistic experts, as depicted in Figure 1. The CEDCC addresses key limitations in prior

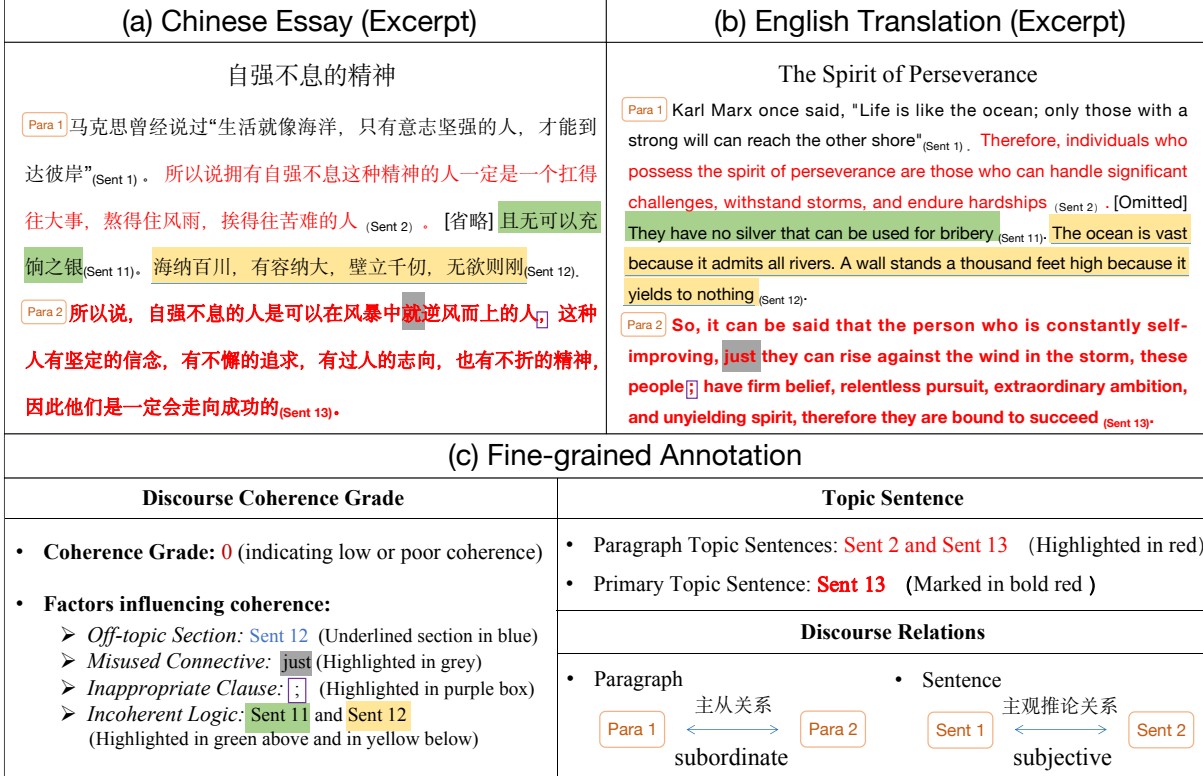

| (a) Chinese Essay (Excerpt) | (b) English Translation (Excerpt) |
|---|---|

自强不息的精神

[Para 1] 马克思曾经说过"生活就像海洋，只有意志坚强的人，才能到达彼岸"(Sent 1)。所以说拥有自强不息这种精神的人一定是一个打得往大事，熬得住风雨，挨得往苦难的人 (Sent 2)。[省略] 且无可以充饷之银(Sent 11)。海纳百川，有容纳大，壁立千仞，无欲则刚(Sent 12)。

[Para 2] 所以说，自强不息的人是可以在风暴中就逆风而上的人，这种人有坚定的信念，有不懈的追求，有过人的志向，也有不折的精神，因此他们是一定会走向成功的(Sent 13)。

The Spirit of Perseverance

[Para 1] Karl Marx once said, "Life is like the ocean; only those with a strong will can reach the other shore"(Sent 1). Therefore, individuals who possess the spirit of perseverance are those who can handle significant challenges, withstand storms, and endure hardships (Sent 2). [Omitted] They have no silver that can be used for bribery (Sent 11). The ocean is vast because it admits all rivers. A wall stands a thousand feet high because it yields to nothing (Sent 12).

[Para 2] So, it can be said that the person who is constantly self-improving, just they can rise against the wind in the storm, these people ; have firm belief, relentless pursuit, extraordinary ambition, and unyielding spirit, therefore they are bound to succeed (Sent 13).

**(c) Fine-grained Annotation**

| Discourse Coherence Grade | Topic Sentence |
|---|---|
| • **Coherence Grade:** 0 (indicating low or poor coherence)
• **Factors influencing coherence:**
➢ *Off-topic Section:* Sent 12 (Underlined section in blue)
➢ *Misused Connective:* just (Highlighted in grey)
➢ *Inappropriate Clause:* ; (Highlighted in purple box)
➢ *Incoherent Logic:* Sent 11 and Sent 12 (Highlighted in green above and in yellow below) | • Paragraph Topic Sentences: Sent 2 and Sent 13 (Highlighted in red)
• Primary Topic Sentence: Sent 13 (Marked in bold red)

**Discourse Relations**
• Paragraph 主从关系 [Para 1] [Para 2] subordinate
• Sentence 主观推论关系 [Sent 1] [Sent 2] subjective |

Figure 1: An example of the CEDCC annotation: (a) Original Chinese Essay (Excerpt), (b) English Translation (Excerpt), and (c) Fine-grained Annotation about discourse coherence.

work: **firstly**, it breaks from the isolationist approach of previous studies. With the integration of `coherence grading`, `topical continuity`, and `discourse relations`, it provides a more comprehensive understanding of discourse coherence. **Secondly**, the detailed annotations, particularly evident in Figure 1(c), capture the nuances of real-world texts. Factors such as off-topic sections, misused connectives, inappropriate clauses, and incoherent logic that influence coherence grade are highlighted. **Finally**, by offering a diverse dataset for Chinese discourse coherence analysis, CEDCC stimulates progress in this area.

Our contributions are summarised as follows:

- We develop the CEDCC, a comprehensive multi-task dataset for discourse coherence assessment, enhancing understanding of Chinese middle school student essays.

- We establish baselines for the CEDCC, setting a reference point for future discourse coherence research.

- Through insightful experiments, we illustrate the impact of coherence on discourse relation recognition and the value of fine-grained annotations for automated essay scoring, encouraging multi-dimensional discourse analysis.

## 2 Related Work

In this section, we delve into three core aspects of discourse coherence. For each, we discuss its objectives, relevant datasets, and their limitations, establishing the backdrop for our proposed dataset.

### 2.1 Discourse Coherence Grading

Discourse Coherence Grading (DCG) in NLP measures text coherence through assessing semantic, structural, and logical aspects. Several studies, such as those using the Grammarly Corpus of Discourse Coherence (GCDC) by Lai and Tetreault (2018), the INSteD dataset for pre-trained language models by Shen et al. (2021), and DDisCo dataset comprising Danish texts by Flansmose Mikkelsen et al. (2022), have contributed significantly to this field by developing specific datasets.

However, these studies often focus on grading overall coherence without pinpointing the specific factors affecting it. Consequently, the derived scores may lack interpretability and offer an incomplete view of discourse coherence. Moreover, most existing resources primarily target English or

other European languages, leaving a noticeable gap in Chinese discourse coherence analysis.

Our CEDCC dataset aims to address these limitations. It predefines common factors impacting discourse coherence, such as *off-topic sections*, *misused connectives*, *inappropriate clauses*, and *incoherent logic*. Besides providing overall coherence scores, our annotators meticulously annotate these factors, thereby improving the interpretability of the coherence grading. Finally, by focusing on Chinese middle school student essays, the CEDCC fills the gap in Chinese discourse coherence evaluation.

## 2.2 Topical Continuity Modelling

Topical continuity modelling aims to quantify the consistency of topics within a document to ensure a cohesive narrative. Several techniques have been proposed for this task. For instance, Amoualian et al. (2017) introduced an LDA-based model to create topic-coherent segments within documents, while Shrivastava et al. (2018) proposed an unsupervised metric to evaluate topic coherence by analyzing latent topic structures. More recently, pre-trained contextualized document embeddings have been used to enhance topic coherence in neural topic models (Bianchi et al., 2021). However, these methods often struggle to capture fine-grained topical variations and require substantial computational resources for processing large volumes of text.

In contrast, our CEDCC dataset addresses this issue by providing granular, sentence-level annotations for both paragraph and overall topics. These detailed annotations capture the nuanced topical variations within documents, adding depth to the task of topical continuity modelling. Furthermore, the focus on these finer details reduces the reliance on intensive pre-processing and segmentation techniques, which often limit the effectiveness of current models. Hence, the CEDCC dataset can serve as a valuable tool for advancing research in topical continuity modelling, especially for tasks that require an understanding of finer topical variations within a document.

## 2.3 Discourse Relation Recognition

Discourse Relation Recognition (DRR) is a significant area in NLP that seeks to identify and classify the relationships between text segments. Numerous efforts, such as the works by Zhou et al. (2022) and Wu et al. (2023), have greatly advanced the DRR field. Given the intimate connection between discourse relations and coherence, researchers have proposed using DRR for coherence assessment, as exemplified by the works of Lin et al. (2011) and Feng et al. (2014), and the investigation into concurrent explicit and implicit relations by Rohde et al. (2018).

However, due to the absence of datasets featuring both discourse coherence grading and relations annotation, prior methods often resort to sentence ordering tasks for coherence assessment (Mohammadi et al., 2020). These tasks involve juxtaposing well-structured text with randomly arranged sentences, which, despite their usefulness, may not fully encapsulate the intricacies of real-world text coherence. To rectify this limitation, our CEDCC dataset includes both discourse relations annotations and coherence scores. This integrated approach offers a more nuanced understanding of how discourse relations contribute to overall text coherence, addressing the deficiencies of previous methods.

## 3 Corpus Construction

This section delineates the process of collection and annotation for the Chinese Essay Discourse Coherence Corpus (CEDCC), designed for extensive discourse coherence analysis.

## 3.1 Data Collection

For the construction of the CEDCC, we collected 501 essays from secondary school students' exam compositions and daily practice. These essays, ranging from 603 to 1,600 tokens with an average of approximately 713.18 tokens, were meticulously selected based on criteria such as genre and teacher-assigned scores. As depicted in Figure 2(a), our dataset spans eight distinct genres, and the distribution of teacher-provided scores is illustrated in Figure 2(b).

We specifically chose secondary school compositions for their significance in discourse coherence research. These essays offer genuine instances of both coherent and incoherent discourse combined with formal language usage. The teacher-assigned scores further enable a correlation between discourse coherence ratings and the overall essay evaluation, addressing a primary challenge in Chinese automated essay scoring which often struggles to integrate discourse coherence due to the absence of appropriate datasets like CEDCC.

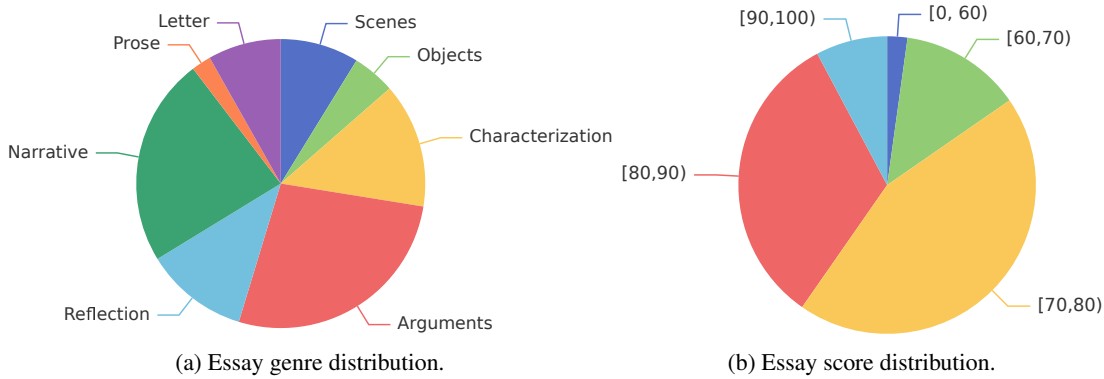

Figure 2: (a) displays the distribution of the 501 essays used to construct the dataset by genre, covering a total of 8 essay genres. (b) shows the distribution of the essays used for annotation in terms of score.

## 3.2 Fine-grained Annotation Format

For each essay in our corpus, our annotation contains three components, *i.e.*, `discourse coherence grade`, `topic sentences` and `discourse relations`.

**Discourse Coherence Grade** In our dataset, each essay—comprising its title and main text—undergoes a coherence grading process. Adhering to conventions from prior studies (Lai and Tetreault, 2018; Flansmose Mikkelsen et al., 2022), annotators use a three-tier coherence grading system: excellent (scored 2), average (scored 1), or poor (scored 0). They also identify and annotate specific issues affecting coherence, such as off-topic sections, misused connectives, and instances of illogical flow. Given the multifaceted nature of discourse coherence, we are contemplating adopting a broader grading scale, possibly a Likert scale (Jebb et al., 2021), in future iterations.

**Topic Sentences** Essays in our dataset are annotated with topic sentences that capture the main themes of their paragraphs, with a primary topic sentence signifying the essay's overarching theme. These can be located anywhere within a paragraph or inferred if explicitly absent.

In contrast to traditional topic continuity models, which emphasize topic words, we prioritize topic sentences for their richer thematic context. This approach suits the varied structures and styles of secondary school essays and has proven effective, as detailed in 4.3.2.

**Discourse Relations** Informed by resources such as the Rhetorical Structure Theory (Mann and Thompson, 1988) and the Penn Discourse Tree-Bank (Prasad et al., 2008), our annotation scheme classifies discourse relations into explicit and implicit relations among paragraph and sentence pairs, based on the presence or absence of connectives.

Annotators identified explicit connectives, pinpointed their arguments, and labeled the discourse relation for explicit relations. For relations without explicit connectives, they inferred implicit relations. Our annotation scheme is tailored to accommodate the unique characteristics of Chinese discourse, such as complex sentence structures and common sentence grouping practices. For a detailed overview of our discourse relation annotation scheme, please refer to Appendix A.

## 3.3 Annotation Process

Our annotation team, consisting of language students and expert reviewers, underwent a training session before starting the annotation process. The dataset was divided into five groups for efficient and consistent annotation. The whole process, involving grading discourse coherence, identifying topic sentences, and defining discourse relations, took three months, resulting in a total of 501 annotated essays. For a detailed overview of our annotation process, please refer to Appendix B.

## 3.4 Data Statistics

We present an overview of our dataset's main characteristics, spanning coherence grades, topic sentences, and discourse relations. The distribution of coherence grades, including Excellent, Average, and Poor, across each genre, is depicted in Table 1. We also detail the frequency and distribution of specific issues impacting coherence in Table 2.

Each essay averages 6 to 8 primary topic sentences. Considering essays typically have 7-8 paragraphs with 5-8 sentences each, we identified a

main topic sentence for every paragraph. While most of these topic sentences start their respective paragraphs, the overall distribution is more diverse, as detailed in Table 3.

Finally, we examine discourse relations in our corpus, considering both those between adjacent paragraphs and within sentences. The distribution of these explicit and implicit relations is presented in Table 4. These relations are further classified into thirteen fine-grained relations, which correspond to four coarse-grained categories, as detailed in Figure 4.

### 3.5 Inner Annotator Agreements

To ensure the annotation quality, Inter-Annotator Agreement (IAA) (Kremer et al., 2014) was measured across various tasks associated with identifying off-topic information, misuse of connectives, inappropriate clause structures, illogical flow in text, sentence connectives and relations, as well as paragraph connectives and relations. Table 5 shows the IAA scores for these tasks across different annotation batches. Further details on IAA calculation can be found in Appendix C.

| Genre | Coherence Grade (%) | | |
|---|---|---|---|
| | Excellent | Average | Poor |
| Scenes | 29.55 | 20.45 | 50.00 |
| Objects | 33.33 | 29.17 | 37.50 |
| Characterization | 37.14 | 30.00 | 32.86 |
| Arguments | 48.89 | 31.11 | 20.00 |
| Reflection | 44.83 | 27.59 | 27.59 |
| Narrative | 37.61 | 30.77 | 31.62 |
| Prose | 27.27 | 36.36 | 36.36 |
| Letter | 50.00 | 19.05 | 30.95 |
| Total | 41.52 | 28.74 | 30.14 |

Table 1: Distribution of coherence grades across different genres, presented as percentages.

| Factor | Count | Avg/Essay | % of Total |
|---|---|---|---|
| Off-Topic | 43 | 0.09 | 2.40% |
| Misused Conn. | 237 | 0.47 | 13.21% |
| Inapp. Clauses | 1244 | 2.48 | 69.34% |
| Incoherent Logic | 270 | 0.54 | 15.05% |
| **Total** | **1794** | **3.58** | **100.00%** |

Table 2: Distribution and average occurrences per essay of factors impacting coherence.

### 3.6 Ethical Issues

All data annotators and expert reviewers were compensated for their contributions. Furthermore, we have acquired explicit permission from both the authors of the essays and their guardians to use the essays for annotation and publication purposes. To safeguard the privacy of the students, all essays in the dataset have been anonymized, ensuring no personal identifiers are present. We deeply appreciate the trust and support shown by all involved parties.

## 4 Experiments

### 4.1 Tasks

Our annotated dataset serves as the foundation for three core tasks, each delving into distinct facets of discourse coherence:

- Discourse Coherence Grading: This task assesses the overall coherence of essays, providing a holistic view of the textual flow and structure.

- Topic Sentence Extraction: This task homes in on the crux of each paragraph, pinpointing the primary sentence that best embodies the central theme. It operates at a micro-level, spotlighting the nuances that contribute to the broader coherence.

- Discourse Relation Recognition: This task is dedicated to unveiling the semantic and logical relations both within sentences and among them. It's instrumental in understanding the intricate interconnections that underpin the textual coherence.

Each task, while distinct, contributes to our understanding of discourse coherence. The global

| Genre | In Paragraph (%) | | | In Full-Text (%) | | |
|---|---|---|---|---|---|---|
| | B | M | E | B | M | E |
| Scenes | 75.71 | 5.31 | 18.98 | 52.27 | 22.73 | 25.00 |
| Objects | 75.95 | 5.70 | 18.35 | 33.33 | 8.33 | 58.33 |
| Characterization | 78.98 | 4.08 | 16.94 | 50.00 | 7.14 | 42.86 |
| Arguments | 75.71 | 5.31 | 18.98 | 41.48 | 25.93 | 32.59 |
| Reflection | 73.96 | 4.43 | 21.61 | 53.45 | 13.79 | 32.76 |
| Narrative | 69.43 | 10.14 | 20.42 | 36.75 | 10.26 | 52.99 |
| Prose | 76.92 | 6.41 | 16.67 | 18.18 | 63.64 | 18.18 |
| Letter | 73.40 | 6.74 | 19.86 | 35.71 | 40.48 | 23.81 |
| Total | 74.40 | 6.23 | 19.37 | 42.51 | 19.62 | 38.32 |

Table 3: Distribution of topic sentences' positions within paragraphs and full text, by genre. B, M, E represent beginning, middle, and end respectively.

coherence of an essay, for example, is influenced by its topic transitions and interwoven discourse relations. Topic sentence extraction, in turn, sheds light on both macro-level coherence and the underlying discourse relations. Together, these tasks position our dataset as a robust tool for probing discourse coherence, paving the path for advancements in NLP and pedagogical research.

| Fine-grained | Paragraph | | Sentence | |
|---|---|---|---|---|
| | Exp. | Imp. | Exp. | Imp. |
| Coherence | 47 | 279 | 197 | 375 |
| Sequence | 73 | 751 | 177 | 1718 |
| Progression | 45 | 130 | 256 | 269 |
| Contrast | 7 | 12 | 67 | 74 |
| Concession | 9 | 7 | 98 | 9 |
| Turnabout | 75 | 34 | 1096 | 83 |
| Refinement | 22 | 427 | 105 | 1651 |
| Generalization | 49 | 618 | 57 | 754 |
| Background | 5 | 127 | 12 | 407 |
| Objective Causal | 11 | 15 | 309 | 172 |
| Subjective Inference | 4 | 1 | 37 | 38 |
| Specific Conditional | 0 | 0 | 121 | 7 |
| Hypothetical Conditional | 3 | 2 | 108 | 9 |
| **Total** | 350 | 2403 | 2640 | 5566 |

Table 4: Distribution of Discourse Relations at Paragraph and Sentence Levels.

## 4.2 Baselines and Evaluation Metrics

We test several existing models on each of our discourse assessment tasks. These models comprise traditional NLP models, pre-trained transformer models such as BERT(Devlin et al., 2019) and RoBERTa(Liu et al., 2019), as well as large-scale language models like ChatGPT, specifically the gpt-3.5-turbo version[1]. We employ both zero-shot and few-shot learning for all tasks. The prompt and details for utilizing ChatGPT can be found in Appendix D.

**Discourse Coherence Grading:** In line with the work of (Lai and Tetreault, 2018), we test a variety of models: Entity-based Models (EGRID (Barzilay and Lapata, 2005) and EGRAPH (Guinaudeau and Strube, 2013)) and neural network models (CLIQUE (Li and Jurafsky, 2017), SENTAVG and PARSEQ (Lai and Tetreault, 2018) ) have shown promise in previous research for this task, and we further include the aforementioned pre-trained and large-scale language models.

**Topic Sentence Extraction:** Inspired by methods from the extractive summarization field, we test models of different types: LEAD-3 and OR-

ACLE are popular summarization baselines; TextRank(Mihalcea and Tarau, 2004) and LDA(Blei et al., 2003) as unsupervised methods; transformer-based models like BERT-ext and BERT-abs, which apply the extractive and abstractive summarization techniques, respectively, based on the code[2] from BERTSum(Liu and Lapata, 2019). Additionally, we include BART(Lewis et al., 2020), T5(Raffel et al., 2020), and ChatGPT, which employ end-to-end generation methods.

**Discourse Relation Recognition:** For this task, we focus on a variety of pre-trained models including BERT, RoBERTa, XLNet(Yang et al., 2019), and ChatGPT, which have shown success in natural language understanding in prior work.

For the evaluation, we employ metrics tailored to each task. For both the Discourse Coherence Grading and Discourse Relation Recognition tasks, the models' performance is assessed using Precision ($P$), Recall ($R$), $F_1$-score ($F_1$), and Accuracy ($Acc$). For the Topic Sentence Extraction task, we use ROUGE-1($R1$), ROUGE-2($R2$) and ROUGE-L ($RL$,$n$-grams overlap measures), BLEU (another $n$-grams overlap measure considering up to 4-grams), and BERTScore(Zhang et al., 2020) (a word overlap measure based on contextual BERT embeddings). This approach allows for a task-focused examination of baselines and metrics, providing clear distinctions between models and the rationale behind their selection and evaluation.

| Batch | Off-Topic | Misused Conn. | Inapp. clauses | Illogical Flow | Primary Topic | Sent. Relations | Para. Topic | Para. Relations |
|---|---|---|---|---|---|---|---|---|
| 0 | 67.27 | 61.71 | 44.46 | 59.37 | 74.39 | 93.10 | 80.84 | 93.88 |
| 1 | 67.61 | 61.76 | 52.82 | 56.97 | 80.94 | 95.24 | 84.64 | 95.36 |
| 2 | 65.89 | 66.71 | 49.22 | 65.69 | 79.26 | 93.26 | 84.35 | 92.23 |
| 3 | 65.32 | 71.99 | 57.41 | 79.39 | 78.94 | 93.59 | 79.35 | 92.76 |
| 4 | 70.41 | 87.11 | 57.26 | 64.51 | 80.97 | 96.13 | 87.41 | 97.40 |
| Avg. | 67.30 | 69.74 | 52.23 | 65.19 | 78.90 | 94.26 | 83.32 | 94.33 |

Table 5: Consistency analysis results showing the inter-annotator agreement (IAA) scores (in percentage) across different aspects of text analysis for various data submission batches (each batch represents a round of annotations). The last row shows the average IAA scores for all batches.

## 4.3 Main Results and Analysis

In this section, we present and analyse the results of the benchmark model for each subtask in turn. We use the annotations from the first 401 compositions in the dataset as the training set, and the annotations from the last 100 compositions as the test set.

---

[1]https://openai.com/blog/chatgpt

[2]https://github.com/nlpyang/PreSumm

### 4.3.1 Discourse Coherence Grading

Table 6 showcases the performance of diverse models for the `Discourse Coherence Grading` task. The Transformer-based models BERT and RoBERTa, especially the latter, displayed superior performance in distinguishing discourse coherence, reflecting their effective utilization of contextual information in the text.

| Model | P(%) | R(%) | $F_1$(%) | Acc(%) |
|---|---|---|---|---|
| EGRID | 35.83 | 35.12 | 33.92 | 40.00 |
| EGRAPH | 27.89 | 28.36 | 27.92 | 43.00 |
| CLIQUE | 33.29 | 33.86 | 28.45 | 29.57 |
| SENTAVG | 33.9 | 34.2 | 33.99 | 45.00 |
| PARSEQ | 35.16 | 33.58 | 31.26 | 34.00 |
| BERT | 41.42 | 36.78 | 34.01 | 57.00 |
| RoBERTa | **50.37** | **47.16** | **46.73** | **58.00** |
| ChatGPT | 34.13 | 30.26 | 31.15 | 55.00 |
| ChatGPT$_{3-shot}$ | 41.37 | 42.40 | 41.49 | 54.00 |

Table 6: Comparative performance of different models for Discourse Coherence Grading.

Entity-based models, namely EGRID and EGRAPH, scored lower than Transformer-based models but still managed to compete with Neural Network models such as SENTAVG, CLIQUE, and PARSEQ. This indicates that even simpler approaches like entity transition patterns hold significant importance in determining discourse coherence. The Large-scale Language Models, represented by ChatGPT and ChatGPT$_{3-shot}$, showed commendable performance, with ChatGPT$_{3-shot}$ notably outperforming its zero-shot counterpart.

### 4.3.2 Topic Sentence Extraction

As Table 7 illustrates, the ORACLE model, due to its utilization of ground truth labels, unsurprisingly achieves the highest scores across all metrics. Among Transformer-based models, BART and T5 outperform others, demonstrating their effectiveness in the topic sentence extraction task. Interestingly, the less complex models, LEAD-3 and LDA, also yield competitive performance. This finding suggests the significant roles of initial sentences and topic coherency for this particular task.

In contrast, when considering pre-trained language models, it is found that ChatGPT and its few-shot variant perform less optimally compared to other models. Despite the remarkable language understanding capabilities of ChatGPT, the model appears to lack a full comprehension of our topic sentence extraction task, despite a detailed intro-

duction given in the prompt. However, its performance notably improves when provided with several examples, indicating the potential benefits of example-based fine-tuning in enhancing its task comprehension.

| Model | R1 | R2 | RL | BLEU | BERTScore |
|---|---|---|---|---|---|
| LEAD-3 | 67.58 | 62.06 | 63.79 | 43.00 | 85.33 |
| ORACLE | **93.91** | **92.78** | **93.45** | **91.52** | **97.16** |
| TextRank | 60.29 | 54.4 | 59.34 | 51.32 | 81.25 |
| LDA | 80.62 | 76.61 | 79.79 | 74.86 | 90.48 |
| BERT-ext | 70.78 | 64.03 | 69.62 | 66.25 | 85.49 |
| BERT-abs | 54.52 | 42.46 | 48.89 | 39.78 | 76.87 |
| BART | 74.58 | 68.9 | 73.32 | 63.51 | 87.51 |
| T5 | 74.75 | 70.33 | 69.94 | 39.33 | 86.15 |
| ChatGPT | 24.74 | 13.06 | 22.8 | 8.29 | 63.22 |
| ChatGPT$_{3-shot}$ | 41.05 | 30.65 | 38.58 | 25.07 | 70.92 |

Table 7: Performance (in %) of various models on Topic Sentence Extraction.

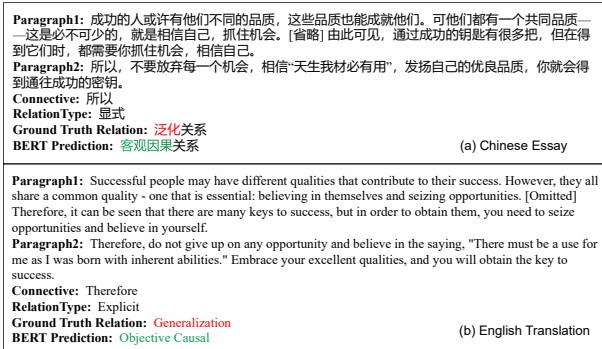

Figure 3: Case study on explict discourse relation recognition among paragraphs

### 4.3.3 Discourse Relation Recognition

Tables 8 and 9 delineate the varied model performance in discourse relation recognition. At the paragraph level, BERT$_{large}$ generally outperforms, highlighting its aptitude in complex contexts, yet certain categories are better addressed by RoBERTa$_{large}$ and XLNet$_{mid}$, underscoring the potential for specialized optimizations. On the sentence-level, BERT models take the lead, but RoBERTa$_{large}$ exhibits prowess in Implicit *Elaboration* and *Causal* relations, emphasizing its nuanced relation handling.

An intriguing observation during inter-paragraph experiments was the underperformance of explicit relations compared to implicit ones, despite the presence of cues like *so*. Delving into this, our examination (Figure 3) revealed that the cue *therefore*, typically signaling causality in inter-sentence

| | Elaboration | | Reversal | | Causal | | Co-occurrence | | Total $F_1$ | | Total $Acc$ | |
|---|---|---|---|---|---|---|---|---|---|---|---|---|
| | Exp. | Imp. | Exp. | Imp. | Exp. | Imp. | Exp. | Imp. | Exp. | Imp. | Exp. | Imp. |
| BERT | 0 | 7.41 | 70.97 | **26.67** | 51.85 | 69.68 | 71.05 | 64.52 | 48.47 | 42.07 | 62.65 | 64.84 |
| BERT$_{large}$ | **44.44** | 24.39 | **78.79** | 13.33 | **73.68** | 71.46 | **77.61** | 62.13 | **68.63** | 42.83 | **74.70** | 64.62 |
| RoBERTa | 0 | 0 | 0 | 0 | 66.67 | 75.85 | 75.29 | 74.65 | 35.49 | 37.63 | 61.45 | 72.53 |
| RoBERTa$_{large}$ | 0 | **28.57** | 70.97 | 12.5 | 68 | **76.04** | 75.95 | **75.29** | 53.73 | 48.10 | 69.88 | **72.75** |
| XLNet | 0 | 7.69 | 41.67 | 0 | 61.02 | 75.54 | 76.92 | 73.22 | 44.90 | 39.11 | 63.86 | 71.65 |
| XLNet$_{mid}$ | 25 | 16.67 | 70.97 | 26.67 | 72.34 | 75.12 | 77.50 | 75.27 | 61.45 | **48.43** | 72.29 | 72.09 |
| ChatGPT | 0 | 6.06 | 47.06 | 5.19 | 34.48 | 12.71 | 67.35 | 60.99 | 37.22 | 21.24 | 55.42 | 41.76 |
| ChatGPT$_{3-shot}$ | 0 | 10.26 | 41.38 | 6.67 | 21.26 | 29.47 | 57.45 | 58.99 | 30.11 | 26.35 | 44.58 | 45.93 |

Table 8: Performance of various models on the Coarse-grained Discourse Relation Recognition between paragraphs. Displayed are the $F_1$ scores (%) for explicit (Exp.) and implicit (Imp.) relations for each type on the CEDCC dataset, and overall $F_1$ scores and $Acc$ for all Level-1 senses.

| | Elaboration | | Reversal | | Causal | | Co-occurrence | | Total $F_1$ | | Total $Acc$ | |
|---|---|---|---|---|---|---|---|---|---|---|---|---|
| | Exp. | Imp. | Exp. | Imp. | Exp. | Imp. | Exp. | Imp. | Exp. | Imp. | Exp. | Imp. |
| BERT | 87.87 | 38.14 | 92.39 | 12.9 | 76.92 | 73.82 | 79.87 | 61.66 | **84.26** | 46.63 | **86.76** | 64.29 |
| BERT$_{large}$ | **87.95** | 38.58 | **92.47** | 6.67 | **77.37** | 74.26 | 79.03 | 64.44 | 84.20 | 45.99 | 86.61 | 65.97 |
| RoBERTa | 85.71 | 29.3 | 92.12 | 8.33 | 72.73 | 73.93 | 80.00 | 63.69 | 82.64 | 43.81 | 85.57 | 65.58 |
| RoBERTa$_{large}$ | 81.51 | **42.33** | 91.98 | 8.33 | 68.71 | **77.95** | **80.49** | 63.87 | 80.67 | **75.25** | 68.15 | 40.48 |
| XLNet | 84.85 | 35.71 | 92.20 | 9.52 | 73.42 | 76.51 | 78.77 | 64.53 | 82.31 | 46.57 | 85.12 | 67.96 |
| XLNet$_{mid}$ | 85.15 | 36.18 | 92.15 | **18.75** | 75.18 | 76.97 | 78.29 | **66.84** | 82.69 | 49.68 | 85.42 | **68.15** |
| ChatGPT | 20.32 | 8.82 | 40.72 | 2.41 | 13.33 | 8.74 | 41.56 | 48.61 | 28.99 | 17.15 | 36.76 | 29.37 |
| ChatGPT$_{3-shot}$ | 38.68 | 10.57 | 59.84 | 8.14 | 33.03 | 31.91 | 48.41 | 46.81 | 44.99 | 24.36 | 49.55 | 34.42 |

Table 9: Performance of various models on the Coarse-grained Discourse Relation Recognition between sentences. Displayed are the $F_1$ scores (%) for explicit (Exp.) and implicit (Imp.) relations for each type on the CEDCC dataset, and overall $F_1$ scores and $Acc$ for all Level-1 senses.

contexts, often prefaced a summary or generalized statement at the paragraph level, thus indicating generalization more than causation. This reflects the nuanced semantic intricacies between paragraphs and the associated modeling challenges, suggesting a refinement in future recognition work to accommodate these variances.

ChatGPT variants show moderate results, suggesting enhancement opportunities. Conclusively, no single model excels consistently, pointing towards the merit of hybrid or specialized strategies. Further granular recognition details are available in Appendix E.

## 5 Discussion

### 5.1 The Impact of Coherence on Discourse Relation Recognition

According to Table 10, while the performances vary across different coherence levels, there are some noticeable patterns. For instance, XLNet consistently achieves its best overall $Acc$ and $F_1$ scores

| Model | Grade | Explicit | | Implicit | | Overall | |
|---|---|---|---|---|---|---|---|
| | | $Acc$ | $F_1$ | $Acc$ | $F_1$ | $Acc$ | $F_1$ |
| BERT | 0 | 55.56 | **43.83** | 55.26 | 29.41 | 55.32 | 39.11 |
| | 1 | **90.91** | 67.73 | 64.08 | 32.93 | **66.67** | **52.60** |
| | 2 | 61.11 | 47.03 | 67.39 | 40.91 | 66.36 | 49.58 |
| RoBERTa | 0 | **66.67** | 38.68 | **72.37** | 37.78 | **71.28** | 38.02 |
| | 1 | 54.55 | 32.05 | 66.99 | **34.66** | 65.79 | 34.40 |
| | 2 | 61.11 | 35.23 | **74.64** | 38.65 | **72.42** | 38.22 |
| XLNet | 0 | 61.11 | 43.48 | 71.05 | **45.12** | 69.15 | **48.98** |
| | 1 | 45.45 | 33.33 | 66.99 | 34.45 | 64.91 | 41.81 |
| | 2 | **68.52** | 52.32 | 73.19 | 47.24 | 72.42 | 57.29 |

Table 10: Performance of different models on discourse relation recognition tasks at different coherence levels.

for essays with the highest coherence level. This suggests that for certain models, essays with better coherence might facilitate improved discourse relation recognition. However, for models like BERT and RoBERTa, the relationship between coherence and performance is less straightforward, indicating further nuances in how text coherence might influence discourse recognition across models.

This outcome underscores the importance of

considering text coherence in the application of discourse relation recognition models, hinting at possible enhancements from employing coherence grades, provided they are obtainable. Additionally, the need for more robust models capable of dealing with text of varying coherence levels is evident, a crucial requirement considering the wide-ranging coherence grades present in real-world data.

Interestingly, the recognition of implicit relations seems to be less sensitive to the coherence grade, highlighting the inherent difficulty of inferring these relations. This underscores the need for more advanced techniques capable of identifying implicit discourse relations, which could significantly enhance the overall performance of discourse relation recognition models.

## 5.2 The Impact of Fine-grained Annotations on Discourse Coherence Grading

| Model | $P(\%)$ | $R(\%)$ | $F_1(\%)$ | $Acc(\%)$ |
|---|---|---|---|---|
| ChatGPT | 34.13 | 30.26 | 31.15 | 55.00 |
| ChatGPT$_{3-shot}$ | **41.37** | 42.40 | **41.49** | 54.00 |
| ChatGPT$_{fine-grained\ annotations}$ | 34.73 | **53.40** | 28.60 | **60.00** |

Table 11: Comparative performance of ChatGPT variations in discourse coherence grading task.

As illustrated in Table 11, the inclusion of fine-grained annotations bolsters the ChatGPT model's performance in discourse coherence grading. Specifically, this method improves recall substantially (53.4%), indicating better detection of various error categories in essays.

Furthermore, this approach yields the highest accuracy (60%) among the compared models, validating the value of fine-grained annotations. These findings reveal that even with infrequent error categories, weakly supervised methods can utilize fine-grained annotations effectively for enhanced performance, underlining their utility in such context-dependent tasks.

## 6 Conclusion

In this paper, we introduced the Chinese Essay Discourse Coherence Corpus (CEDCC), a richly annotated, diverse dataset aiming to address limitations in current discourse coherence research. Our dataset integrates aspects of coherence grading, topical continuity, and discourse relations, thereby breaking away from the isolated approach of previous studies. We further demonstrated the importance of fine-grained annotations and the role of

text coherence in discourse relation recognition. Our findings set a benchmark for future discourse analysis research in Chinese, potentially sparking further advancements in the field.

## Limitations

The limitations of our corpus include:

- **Limited by data scale**: While our dataset is among the largest in this field, its size is still constrained. The diversity and complexity of discourse coherence phenomena imply that the larger the dataset, the more comprehensive its coverage of these phenomena. Consequently, the current size of our dataset might limit the performance and generalizability of models trained on it.

- **Constraints of manual annotation**: Our dataset relies significantly on manual annotations by linguistic experts. Nonetheless, due to the labor-intensive and time-consuming nature of this process, there are inevitable limitations on the volume of annotated data. Further, the inherent subjectivity of manual annotation might lead to potential inconsistencies and bias in the annotated labels.

- **Dependent on essay quality**: Our dataset consists of essays authored by middle school students, which might contain language inaccuracies or logical inconsistencies absent in more formal, polished texts. These could affect the learning and generalization capabilities of the models, particularly when applied to other types of text such as academic or formal writing.

## Acknowledgement

We appreciate the support from National Natural Science Foundation of China with the Main Research Project on Machine Behavior and Human-Machine Collaborated Decision Making Methodology (72192820 & 72192824), Pudong New Area Science & Technology Development Fund (PKX2021-R05), Science and Technology Commission of Shanghai Municipality (22DZ2229004), Shanghai Trusted Industry Internet Software Collaborative Innovation Center and East China Normal University International Conference Grant Programme.

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

# A    Annotation Scheme of Relations

Our annotation scheme, inspired by Rhetorical Structure Theory (RST), Penn Discourse Treebank (PDTB), Chinese compound sentence theory, and Chinese sentence group theory (MANN and THOMPSON, 1988; Prasad et al., 2008), is intricately tailored to the nuances of Chinese discourse. It encompasses four tiers and thirteen labels, capturing a broad spectrum of logical semantic types in Chinese discourse. This versatility allows for a unified analysis across diverse discourse units.

While our scheme might not explicitly differentiate arguments in the manner of the PDTB, it imbues directionality in discourse relations. Unlike the traditional undirected relations, our model delineates both parallel and subordinate relationships among discourse units:

- **Parallel Relations**: These involve discourse units of equal standing, without any hierarchical distinction. Examples include relations like Co-occurrence and Elaboration. In cases such as coherence or contrast, the order of discourse units can often be changed without affecting the overall meaning.

- **Subordinate Relations**: These denote a hierarchy, where one discourse unit takes precedence over the other. Relations like Reversal and Causal are illustrative of this category, where the primary discourse unit can be seen as the main clause, and the subsequent as the subordinate clause.

This distinction ensures our scheme neither oversimplifies nor overcomplicates the importance of each discourse relation. It strikes an optimal balance: intricate enough to convey the depth of Chinese discourse yet straightforward enough to minimize potential annotation challenges and subjectivity.

Figure 4 visually illustrates our schema, emphasizing the interplay between coarse-grained and fine-grained discourse relation labels. Detailed definitions of these fine-grained labels follow:

- **Coherence (Co-occurrence)**: This label describes aspects of the same event, related events, or contrasting situations that coexist, co-occur, or oppose in meaning. These aspects can be reordered without altering the overall sentence meaning.

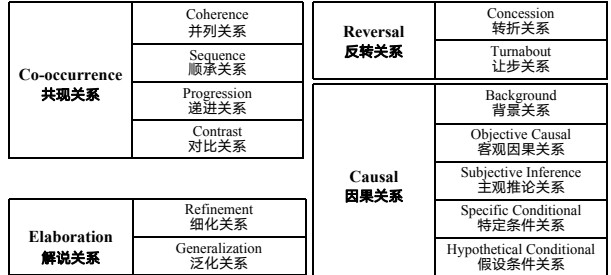

Figure 4: Hierarchical representation of the annotation scheme for coarse and fine-grained discourse relations.

- **Sequence (Co-occurrence)**: This label is used when there is a temporal, spatial, procedural, or logical sequence of events within the discourse units of a chapter. It includes both sequential and inverse orders but does not encompass simultaneous events.

- **Progression (Co-occurrence)**: This label applies when a subsequent discourse unit represents an advance in quantity, quality, scope, or time from the preceding discourse unit, indicating increased depth. The order of the discourse units is usually non-interchangeable.

- **Contrast (Co-occurrence)**: This label represents a notable, direct, and relative difference or similarity between entities, concepts, ideas, actions, states, etc., appearing in the text.

- **Concession (Reversal)**: This label is used when one discourse unit presents a hypothetical scenario, and another unit describes a situation that contradicts or opposes the hypothesis.

- **Turnabout (Reversal)**: This label applies when one discourse unit presents an objective fact, and another unit recounts a situation that opposes or contradicts it.

- **Refinement (Elaboration)**: This label is used when a subsequent discourse unit refines a prior unit, including examples, explanations, illustrations, additions, etc. Reversing the order of the units would transform the relationship into a generalization.

- **Generalization (Elaboration)**: This label applies when a subsequent discourse unit generalizes, summarizes, or extends a prior unit. If the units were reversed, the relationship would be considered a refinement.

- **Background (Causal)**: This label describes the introduction of events, places, histories, etc., which often occurs in chapters and forms a contextual relationship with the main body of the chapter.

- **Objective Causal (Causal)**: This label is used when one discourse unit states a cause, and another unit states the objective result that ensues from the cause.

- **Subjective Inference (Causal)**: This label applies when one discourse unit states the factual basis, and another unit states the subjective conclusion inferred from it.

- **Specific Conditional (Causal)**: This label is used when one discourse unit presents a specific condition, and another unit states the result inferred from that condition.

- **Hypothetical Conditional (Causal)**: This label is used when one discourse unit presents a hypothetical condition, and another unit describes the outcome if the condition were met or the measures needed to meet it.

## B Detailed Annotation Process

Our annotation process was carried out by a team composed of four undergraduates, four postgraduates from language-related fields, and four expert reviewers with experience in Chinese teaching. The principle of minimal changes was followed in the process to retain the original language used by the secondary school students.

Before the actual annotation process, the team underwent a training session to familiarize themselves with the tasks. The tasks included grading the discourse coherence, identifying the topic sentences, and defining discourse relations.

To ensure efficiency and consistency, the data was divided into five groups for annotation. The initial annotation was done by the undergraduate and postgraduate students, while the expert reviewers validated and corrected their work. This process was aimed at maintaining the quality and consistency of the annotations.

Furthermore, we organized weekly online discussions to address any common issues that arose during the annotation process. The discussion also served as a platform to make necessary adjustments in the annotation process.

The entire process spanned three months, during which a total of 501 essays were annotated. This structured approach ensured a streamlined annotation process, resulting in a richly annotated corpus that can facilitate subsequent language model training and research.

## C Inter-Annotator Agreement (IAA) Calculation

In this study, we adopted an Inter-Annotator Agreement (IAA) measure, computed as follows:

$$
\begin{aligned}
\mathbf{PA} &= \frac{1}{|P|} \sum_{(i,j) \in P} \mathbf{PA}_{ij} \\
\mathbf{PA}_{ij} &= \sum_{k=1}^{N} \frac{1}{N} \frac{\left|s_k^i \cap s_k^j\right|}{\left|s_k^i \cup s_k^j\right|}
\end{aligned}
\tag{1}
$$

In the above equations, $\mathbf{PA}$ denotes the overall Pairwise Agreement (PA) among the annotators, with $|P|$ representing the total number of annotator pairs. $\mathbf{PA}_{ij}$ represents the agreement for each pair of annotators $i$ and $j$, summed across all $N$ text fragments. $s_k^i$ and $s_k^j$ denote the annotations made by annotator $i$ and $j$ on text fragment $k$, respectively.

Table 5 in the main text presents the IAA scores across different tasks performed during various data batch submissions. Each row corresponds to an individual data batch, with each entry indicating the respective IAA score for the corresponding task. The rigorous calculation of IAA allows us to evaluate the quality and reliability of our annotation process, revealing areas of high agreement and highlighting areas for potential improvement.

## D ChatGPT Prompt Configuration

For all tasks, including `Discourse Coherence Grading`, `Topic Sentence Extraction`, and `Discourse Relation Recognition`, we employ both zero-shot and few-shot learning strategies. Additionally, we discuss how the incorporation of fine-grained annotations enhances the performance of the ChatGPT model in discourse coherence grading task. Please note that the original prompts were written in Chinese. We provide here their English translated versions.

### D.1 Discourse Coherence Grading

The prompts we use for this task are as follows:
In the zero-shot setting, we ask the model:

> Given an essay, you are tasked with assessing its overall coherence and providing a grade. The grade should be 0 for

incoherent, 1 for average, and 2 for excellent coherence. The provided essay will be in the format 'essay_id \t essay_title \t essay_content', where '\t' represents a tab space, and different paragraphs in the essay content are separated by '\n'. Please return the result in the format '"id":[essay_id],"LogicGrade":[grade]'.**[T]**

In the 3-shot setting, the prompt is modified as follows:

> Given an essay, you are tasked with assessing its overall coherence and providing a grade. The grade should be 0 for incoherent, 1 for average, and 2 for excellent coherence. The provided essay will be in the format 'essay_id \t essay_title \t essay_content', where '\t' represents a tab space, and different paragraphs in the essay content are separated by '\n'. Please return the result in the format '"id":[essay_id],"LogicGrade":[grade]'. Here are three sample essays with their coherence grades for reference. The format of these samples is 'essay_id \t essay_title \t essay_content \t coherence_grade', where '\t' represents a tab space.**[S][T]**

Here, **[S]** represents the three provided samples, and **[T]** is the associated information for the essay that needs to be assessed.

### D.2 Topic Sentence Extraction

For the task of topic sentence extraction, we use the following prompts for both zero-shot and 3-shot learning settings:

In the zero-shot setting, we present the model with this prompt:

> Your task is to extract the topic sentence from the following paragraph. The topic sentence should be a complete sentence that summarizes, narrates, and explains the theme of the paragraph. You should select as complete a sentence as possible from the original text. If the paragraph does not have a clear topic sentence, please answer 'There is no topic sentence.' The current paragraph is: **[P]**. Please provide the topic sentence.

In the 3-shot setting, the model is prompted with:

> Your task is to extract the topic sentence from the following paragraph. The topic sentence should be a complete sentence that summarizes, narrates, and explains the theme of the paragraph. You should select as complete a sentence as possible from the original text. If the paragraph does not have a clear topic sentence, please answer 'There is no topic sentence.' Note that only a few paragraphs lack a clear topic sentence. Here are three examples to help you understand the task: **[S]** Now, the paragraph is: **[P]**. Please provide the topic sentence.

Here, **[S]** represents the three provided samples, and **[P]** represents the content of the paragraph that needs to be processed.

### D.3 Discourse Relation Recognition

For the task of discourse relation recognition, we present the model with different prompts in the zero-shot and 3-shot settings:

In the zero-shot setting, the model receives the following prompt:

> Your task is to recognize the discourse relation between two texts. You can find possible discourse relations and their definitions in the following list: **[D]**. The input format is: {"Text1": "First text", "Text2": "Second text"}. For example, if the input is: {"Text1": "I saw Nüwa first killed a giant turtle, propping up the sky with its legs.", "Text2": "Then, she killed a black dragon."}, then you should output: 'Sequential relationship'. **[T]**

In the 3-shot setting, we present the model with the following prompt:

> Your task is to recognize the discourse relation between two texts. You can find possible discourse relations and their definitions in the following list: **[D]**. Now, let's look at some examples: **[S]**. For example, if the input is: {"Text1": "I saw Nüwa first killed a giant turtle, propping up the sky with its legs.", "Text2": "Then, she killed a black dragon."}, then you should output: 'Sequential relationship'. **[T]**

Here, **[D]** provides the detailed definitions of discourse relations (see Appendix A for more details), **[S]** represents the given three examples, and **[T]** represents the text formatted according to the norms.

## D.4 Fine-Grained Annotation in Discourse Coherence Grading

The prompts we use for this task are as follows:

> Your task is to read and understand a middle school essay, and then identify and count the fine-grained factors that impact the coherence of the essay. This includes the number of times the topic sentence deviates, the number of times connectives are improperly used, the number of times sentences are illogically broken, and the number of times the logical relation between contexts does not flow smoothly. The essay is provided in the format 'Article ID\tArticle Title\tText Content', where '\t' denotes a tab space and '\n' indicates a paragraph break. You should return your result in the format '"id":[Article ID],"OffTopic":[Number of Times Topic Sentence Deviates],"MisusedConnectives":[Number of Times Connectives Improperly Used],"InappropriateClauses":[Number of Times Sentences Illogically Broken],"IncoherentLogic":[Number of Times Logical Relation Between Contexts Doesn't Flow Smoothly]'. It's important that you analyze every article on a word-for-word basis and mark all possible errors.
>
> Here is an example of how an essay analysis and returned error count would look like to help you understand the task:**[E][T]**

Here, **[E]** represents the given example and **[T]** represents a well-formatted example of essay analysis and returned error count.

## E Fine-grained Discourse Relation Recognition

As shown in Table 12, for explicit (inter-paragraph) discourse relations, RoBERTa performed the best with an overall $F_1$ score of 16.02%, followed by XLNet at 15.80%, and BERT at 14.20%. In contrast, ChatGPT and ChatGPT$_{3-shot}$ showed relatively poorer performance, scoring 9.48% and 7.00%, respectively. In terms of overall accuracy ($Acc$), XLNet led with a score of 42.17%, closely followed by RoBERTa at 40.96%. However, even under the best circumstances, there remains substantial room for improvement in these models.

In implicit discourse relations, XLNet again performed the best in terms of the overall $F_1$ score, with 19.97%, followed by RoBERTa and BERT with 16.13% and 16.14%, respectively. However, the performance of ChatGPT and ChatGPT$_{3-shot}$ lagged again in this task, scoring 5.66% and 6.57% respectively. In terms of overall accuracy ($Acc$), XLNet led with a score of 55.38%, while the performances of other models were not much different, and ChatGPT and ChatGPT$_{3-shot}$ were still underperforming.

In Table 13, for discourse relations between sentences, the performance of the models is similar to the situation with inter-paragraph relations. XLNet, RoBERTa, and BERT performed well in most cases, while ChatGPT and ChatGPT$_{3-shot}$ performed poorly in most tasks.

In conclusion, although large pre-trained language models have achieved significant results in many NLP tasks, there is still room for improvement in their performance on fine-grained discourse relation recognition tasks, especially in the recognition of implicit discourse relations. Notably, ChatGPT and ChatGPT$_{3-shot}$ generally performed poorly in this task, possibly because their pre-training process lacked training data specifically targeting discourse relations. Therefore, in future work, we may need to pay more attention to specific training techniques for this task to improve model performance.

| | Explict | | | | | Implict | | | | |
|---|---|---|---|---|---|---|---|---|---|---|
| | BERT | RoBERTa | XLNet | ChatGPT | ChatGPT$_{3-shot}$ | BERT | RoBERTa | XLNet | ChatGPT | ChatGPT$_{3-shot}$ |
| Hypothetical Conditional | 0 | 0 | 0 | 0 | 0 | 0 | 0 | 0 | 0 | 0 |
| Progression | 0 | 0 | 0 | **16.67** | 0 | 0 | 0 | **8.33** | 0 | 0 |
| Sequence | 44.44 | **54.55** | 48.28 | 28.17 | 33.90 | 54.20 | 54.40 | **61.73** | 46.05 | 44.55 |
| Subjective Inference | 0 | 0 | 0 | 0 | 0 | 0 | 0 | 0 | 0 | 0 |
| Turnabout | 11.11 | 0 | 30.00 | **43.75** | 37.04 | 0 | 0 | 0 | **2.74** | 8 |
| Coherence | **64.00** | 62.50 | 55.56 | 0 | 0 | 62.02 | 62.41 | **74.29** | 0 | 0 |
| Refinement | 25 | **46.15** | 16.67 | 0 | 0 | 38.15 | **42.31** | 40.52 | 0 | 0 |
| Generalization | 40.00 | 45.00 | **54.90** | 34.78 | 20.00 | 55.44 | 50.58 | **58.71** | 16.56 | 32.85 |
| Concession | 0 | 0 | 0 | 0 | 0 | 0 | 0 | 0 | 0 | 0 |
| Background | 0 | 0 | 0 | 0 | 0 | 0 | 0 | **16.00** | 8.33 | 0 |
| Contrast | 0 | 0 | 0 | 0 | 0 | 0 | 0 | 0 | 0 | 0 |
| Objective Causal | 0 | 0 | 0 | 0 | 0 | 0 | 0 | 0 | 0 | 0 |
| Specific conditional | 0 | 0 | 0 | 0 | 0 | 0 | 0 | 0 | 0 | 0 |
| Total $F_1$ | 14.20 | **16.02** | 15.80 | 9.48 | 7.00 | 16.14 | 16.13 | **19.97** | 5.66 | 6.57 |
| Total $Acc$ | 33.73 | 40.96 | **42.17** | 28.55 | 23.36 | 49.01 | 48.57 | **55.38** | 28.40 | 30.53 |

Table 12: Performance of various models on the Fine-grained Discourse Relation Recognition between paragraphs. Displayed are the $F_1$ scores (%) for Explicit and Implicit relations for each type on the CEDCC dataset, and overall $F_1$ scores and $Acc$ for all Level-2 senses.

| | Explict | | | | | Implict | | | | |
|---|---|---|---|---|---|---|---|---|---|---|
| | BERT | RoBERTa | XLNet | ChatGPT | ChatGPT$_{3-shot}$ | BERT | RoBERTa | XLNet | ChatGPT | ChatGPT$_{3-shot}$ |
| Hypothetical Conditional | 78.95 | 77.33 | **79.49** | 30.19 | 38.81 | 0 | 0 | **20.00** | 0 | 0 |
| Progression | 59.32 | 55.86 | **60.50** | 4.55 | 8.11 | 18.18 | 15.19 | 18.18 | **22.50** | 4.26 |
| Sequence | 43.75 | 56.25 | **57.14** | 11.69 | 12.93 | **54.92** | 54.85 | 54.29 | 38.75 | 37.89 |
| Subjective Inference | **66.67** | 0 | 0 | 0 | 0 | **18.18** | 0 | 0 | 0 | 0 |
| Turnabout | 90.75 | 91.90 | **91.95** | 39.25 | 59.15 | **17.65** | 6.90 | 10.00 | 1.63 | 6.10 |
| Coherence | 64.71 | **70.09** | 61.39 | 18.46 | 39.58 | 50.00 | 45.61 | **52.54** | 13.33 | 10.99 |
| Refinement | **71.26** | 70.59 | 70.45 | 0 | 0 | 66.57 | 66.58 | **70.69** | 0 | 0.56 |
| Generalization | 62.30 | **66.67** | 62.50 | 25.00 | 32.88 | 52.28 | 51.76 | **55.66** | 13.40 | 19.44 |
| Concession | 13.33 | **14.29** | 13.33 | 0 | 11.11 | 0 | 0 | 0 | 0 | **25.00** |
| Background | 0 | 0 | 0 | 0 | 0 | 26.79 | 14.12 | **36.36** | 0 | 2.47 |
| Contrast | 42.86 | 0 | **51.61** | 19.05 | 23.53 | 22.22 | 0 | **40.00** | 7.84 | 9.2 |
| Objective Causal | 87.88 | 86.15 | **91.34** | 0 | 16.90 | **28.57** | 21.28 | 23.26 | 0 | 0 |
| Specific conditional | 78.48 | 75.95 | **81.58** | 4.55 | 20.83 | 0 | 0 | 0 | 0 | 0 |
| Total $F_1$ | **58.48** | 51.16 | 55.48 | 11.75 | 20.29 | 27.34 | 21.25 | **29.31** | 7.50 | 8.92 |
| Total $Acc$ | 77.23 | 77.23 | **77.89** | 21.80 | 36.06 | 53.08 | 52.48 | **56.75** | 20.41 | 18.37 |

Table 13: Performance of various models on the Fine-grained Discourse Relation Recognition between sentences. Displayed are the $F_1$ scores (%) for Explicit and Implicit relations for each type on the CEDCC dataset, and overall $F_1$ scores and $Acc$ for all Level-2 senses.