# OpenReview forum: "A Multi-Task Dataset for Assessing Discourse Coherence in Chinese Essays: Structure, Theme, and Logic Analysis"
_EMNLP/2023/Conference — EMNLP 2023 Main_

### Official Review · Reviewer_i1co · 2023-07-20

**Soundness:** 3

**Excitement:**

3: Ambivalent: It has merits (e.g., it reports state-of-the-art results, the idea is nice), but there are key weaknesses (e.g., it describes incremental work), and it can significantly benefit from another round of revision. However, I won't object to accepting it if my co-reviewers champion it.

**Paper Topic And Main Contributions:**

The paper presents a novel benchmark for assessing discourse coherence in human-generated texts in Chinese language. In particular, the dataset consists of 501 essays of at least 600 words each collected from secondary school students. The authors, together with the novel dataset, introduced a three-fold labelling schema for multi-task setting: discourse coherence grade, that is each essay is provided with a coherence score considering the text as a whole; topic sentences, that is annotators are requested to identify the sentence encapsulating the major claim of each paragraph and the discourse relation, where annotators are requested to identify the relation intervening between paragraph and sentence pairs.
Finally, the authors assessed several models in the three tasks: the Discourse Coherence Grading, where models are requested to provide the global coherence of essays; the Topic Sentence Extraction task identifies the primary sentence in each paragraph, encapsulating the main idea; and the Discourse Relation Recognition, where models are required to detect the semantic and logical relations both within sentences or paragraphs.
The results point out that no single model excels across all tasks and categories, advocating for a hybrid approach that leverages individual model strengths or applying task/category-specific optimizations.
Interestingly, ChatGPT performance is low across all assessed tasks. Additionally, explicit discourse relations in inter-paragraph tasks underperformed compared to their implicit counterparts.

**Questions For The Authors:**

1. Will the dataset be released? I could not find any related statement.
2. Is the score depicted in Figure 2 provided by the teacher? If so it may be beneficial to make this explicit on paragraph 3.1.

**Reasons To Accept:**

The topic is interesting, the paper is well-written and easy to follow. The paper is self-contained. The authors provide clear motivations underlying the work. The paper introduces a novel multi-task benchmark that could be easily plugged in a framework designed to assess Large Language models so as to provide linguistically grounded intuitions on how such models  understand text coherence.

**Reasons To Reject:**

I feel no reason to reject the paper.

**Reproducibility:**

3: Could reproduce the results with some difficulty. The settings of parameters are underspecified or subjectively determined; the training/evaluation data are not widely available.

**Reviewer Confidence:**

3: Pretty sure, but there's a chance I missed something. Although I have a good feel for this area in general, I did not carefully check the paper's details, e.g., the math, experimental design, or novelty.

**Typos Grammar Style And Presentation Improvements:**

Please find some minor comments below:
- please consider anticipating the extension of the acronym NLP, it has been used on line 024 for the first time, but has been extended on line 172;
- please consider rewriting Firstly on line 080 as firstly;
- please consider rewriting four on line 225 as three;
- please consider adding a blank space before round brackets when referring to the literature in case such as for lines 251 (Rhetorical Structure Theory(Mann and
252 Thompson, 1988)) and 253 (Penn Discourse Tree-
253 Bank(Prasad et al., 2008));

---

> ### Author Rebuttal · Authors · 2023-08-26
>
> Response to questions:
>
> 1. Yes, our dataset will be released shortly.
> 2. Indeed, the scores in Figure 2 were provided by the teachers. Thank you for pointing this out; we will make it explicit in the subsequent versions of the paper.
>
> Furthermore, thank you very much for all the suggestions you have provided for our paper.

---

### Official Review · Reviewer_FdLx · 2023-07-31

**Soundness:** 2

**Excitement:**

2: Mediocre: This paper makes marginal contributions (vs non-contemporaneous work), so I would rather not see it in the conference.

**Missing References:**

- On the Creation of a Corpus for Coherence Evaluation of Discursive Units.
  Elham Mohammadi, Timothe Beiko and Leila Kosseim.
  Proceedings of the Twelfth Language Resources and Evaluation Conference, pp.1067-1072, 2020.


**Paper Topic And Main Contributions:**

In this work, the authors took issue with the independence of three kinds of coherence-related research topics (i.e., discourse coherence grading, topic continuity modeling and discourse relation recognition), and they focused on studying those three types of topics simultaneously. More concretely,
the authors first collected a Chinese essay data that consist of 501 essays written by Chinese middle school students, and then conducted three types of coherence-related annotations (discourse coherence grading, topic sentence selection and discourse relation annotation) on the essay data. The authors employed 3-level discourse grading (excellent, average and poor) for annotating the labels of discourse coherence grading. In addition, as for annotating discourse relations, the authors introduced their own discourse relations that consist of four types of coarse-grained relations and thirteen types of fine-grained relations instead of the existing discourse labels such as RST and PDTB. The authors also conducted experiments of automatic discourse coherence grading, topic sentence selection, and discourse relation recognition using the annotated data. For example, as for the evaluation of discourse grading, the authors compared the traditional discourse grading methods using entity grid or entity graph with BERT/RoBERTa-based methods. The authors also utilized ChatGPT to assess the grade of an input essay by giving an appropriate prompt to ChatGPT. In the evaluations of sentence selection, the authors compared the language model-based methods with the traditional methods (e.g., lead-3 and textrank). Finally, in discourse relation recognition, language model-based methods (BERT/RoBERTa/XLNet) were compared with ChatGPT-based methods.


**Questions For The Authors:**

- Question A: According to the explanation in Appendix A, the discourse relations that were introduced in this work seem to be undirected, while the existing discourse relations are typically directed. Why did the authors adopt such undirected relations? I think that such undirected relations would be unsuitable for detailed discourse analysis.

- Question B: I could not find any mention of data release, but do you plan to release the data you have created?


**Reasons To Accept:**

- The dataset created in this work might be useful for researchers who are interested in Chinese discourse analysis.

**Reasons To Reject:**

- The research motivation of this work come from the background in which the three relevant research topics (i.e., discourse coherence grading, topic transition modeling and discourse relation recognition) have been studied independently, but unfortunately, this work also followed the task independence. That is, the created data was only used to evaluate a single research topic without reference to the labels annotated for the other research topics.
In the view of the research motivation, it is rather important to investigate, for example, correlation/dependence of the three types of manually annotated labels to emphasize the usefulness of the created data. However, no such analysis has been done at all in this work. The paper just shows the frequency (or ratio) of the annotated labels from a single coherence-related research topic.

- There is no sufficient explanation as to why sentence-level topic annotation was employed instead of the existing topic continuity modeling. Since topic continuity modeling is different from the sentence selection annotation as described in Section 2.2, it seems unreasonable to annotate the selected topic sentence as a replacement for topic continuity (or transition).

- The authors split the annotated data into the training and test set only. This means that there is no development set, which is normally used to determine hyper-parameters when fine-tuning a pre-trained language model such as BERT and RoBERTa. For this reason, it is difficult to believe that each evaluation was done in a reliable manner.

**Reproducibility:**

2: Would be hard pressed to reproduce the results. The contribution depends on data that are simply not available outside the author's institution or consortium; not enough details are provided.

**Reviewer Confidence:**

4: Quite sure. I tried to check the important points carefully. It's unlikely, though conceivable, that I missed something that should affect my ratings.

---

> ### Author Rebuttal · Authors · 2023-08-26
>
> Thanks for your careful and valuable comments. We will explain your concerns point by point.
>
> Response to reasons for rejection:
> 1. In our study, we indeed conducted an in-depth analysis of the correlations among various research topics affecting discourse coherence. Specifically, in the "Discussion" section, we elaborated on the impact of coherence on discourse relation recognition and the significance of fine-grained annotations for automated essay scoring (L465-L507). Our dataset is multi-task in nature, and for each sub-task, we tested a range of models based on different methodologies and architectures, from rule/entity-based models to the latest generative LLMs. The appendix further discusses the interrelation and dependencies among the sub-tasks. Although our exploration isn't exhaustive, we firmly believe that such a dataset is invaluable for a comprehensive assessment of discourse coherence. Furthermore, this dataset holds significant value for future research, such as delving deeper into the correlations between the sub-tasks.
> 2. Traditional topic continuity modeling primarily relies on annotating topic words. However, our choice to annotate topic sentences is due to the richer information they encompass compared to individual topic words. Furthermore, our dataset can indeed be applied to topic continuity modeling tasks. As shown in Table 7, the representative topic continuity modeling method, LDA, achieves commendable performance on our dataset, second only to the oracle method.
> 3. Thank you for raising the concern regarding the absence of a dev set in our initial submission. Due to constraints related to data volume and the paper submission deadline, we initially reported results on the test set. We recognize the importance of evaluating on a separate validation set for a more objective assessment of model performance. We’re pleased to inform you that we have now constructed a dev set and have conducted evaluations on it. The results from this validation set, along with any relevant updates, will be incorporated into the final version of the paper upon acceptance. We believe that this will provide a more comprehensive and reliable evaluation of our proposed methods.
>
> Response to questions:
>
> 1. Regarding your point about undirected relations, I'd like to clarify. In our study, there's actually a hierarchical relationship between coarse-grained and fine-grained discourse relations, which is indeed directed, as depicted in Table 3. When annotators label the fine-grained relations between argument pairs, they naturally infer the corresponding coarse-grained relations.
> 2. Yes, with the aim of fostering research and further analyses, we plan to release the dataset we've created in the near future.

---

### Official Review · Reviewer_cwRY · 2023-08-07

**Soundness:** 3

**Excitement:**

3: Ambivalent: It has merits (e.g., it reports state-of-the-art results, the idea is nice), but there are key weaknesses (e.g., it describes incremental work), and it can significantly benefit from another round of revision. However, I won't object to accepting it if my co-reviewers champion it.

**Paper Topic And Main Contributions:**

This paper introduces the Chinese Essay Discourse Coherence Corpus (CEDCC), a comprehensive dataset designed for assessing discourse coherence in learners' essays through multiple tasks. Unlike previous research that focused on specific aspects of coherence, the CEDCC addresses this gap by including evaluations in three tasks: discourse coherence grading, topic sentence extraction , and discourse relation recognition.

**Reasons To Accept:**

The main strength of this work is that the dataset is composed of real-world texts, even if the authors ackowledge that the use of texts produced in the context of middle school may suffer for the overall quality of the essays.
Another valuable aspect is that, for each subtask, authors test many models relying on different approaches and architectures, ranging from rule/entity-based models to  recent generative LLMs.

**Reasons To Reject:**

I found many doubtful passages.

The annotation of coherence in 3 classes sounds like limiting the possibility to grade coherence, that is a complex and multifaceted phenomenon. A wider Likert scale would have been more suitable.

In 3.4 Data Statistics, some basic stats, such as size of the corpus/dataset in terms of tokens, average tokens per essay, and so on, may give the reader a more complete idea of the dataset.

In the same section, at 286-289, authors say that for each paragraph there is an average of 7 topic sentences. Earlier in the text, authors wrote that annotators had to identify topic sentences per paragraph, then select the main topic sentence. I wonder how long an average paragraph may be in secondary school essays? Considering the Schaffer method (first example I have in mind, maybe not the best), a well written paragraph in is composed of 8 sentences. In light of this, considering the majority of the sentences in a paragraph as topic sentences is an unnecessary step.

at 341, it would be more correct to cite the correct/actual model used (e.g. code-davinci-002, text-davinci-002, text-davinci-003). ChatGPT is an impelemntation of a chatbot based on those mdoels.

468-474: There’s not a trend in those results. Best performances appear randomly across the three levels

Table 8: Are numbers correct? Majority of the tested models do perform better with implicit relations with respect to explicit. It sounds odd

The formula for the inter annotator agreement at 854 reminds me the Jaccard Index, but as a sort of variant. Please provide references for it.

Finally, I would like to remind the authors that Appendices are not a way to overcome length limitation. As it is stated in the ACL's paper formatting guidelines:
"Appendices are material that can be read, and include lemmas, formulas, proofs, and tables that are not critical to the reading and understanding of the paper."
(for reference: https://acl-org.github.io/ACLPUB/formatting.html#appendices)

Appendices E and F are not critical, but they represent another experiment. The author have to integrate it into the main body of the paper or cut it out.

**Reproducibility:**

3: Could reproduce the results with some difficulty. The settings of parameters are underspecified or subjectively determined; the training/evaluation data are not widely available.

**Reviewer Confidence:**

4: Quite sure. I tried to check the important points carefully. It's unlikely, though conceivable, that I missed something that should affect my ratings.

---

> ### Author Rebuttal · Authors · 2023-08-26
>
> Thank you for your invaluable feedback and thorough review. The paper introduces a novel multi-task benchmark that could be easily plugged in a framework designed to assess Large Language models so as to provide linguistically grounded intuitions on how such models understand text coherence. To address your concerns, I offer the following responses:
> 1. The choice of three levels for coherence grading was made in reference to previous datasets such as GCDC and DDisCo. Considering the complexity and multifaceted nature of discourse coherence, we provided clear definitions and annotations for fine-grained factors under the guidance of linguistic professors. We recognize the merit of the Likert scale and will consider introducing more grading levels in future works.
> 2. Thank you for pointing out the importance of providing more detailed statistics in section 3.4. To address your suggestion: Our dataset comprises 501 essays written by secondary school students. The essay with the maximum number of tokens contains 1,600 tokens, while the one with the minimum has 603 tokens. On average, each essay consists of approximately 713.18 tokens. We commit to providing more comprehensive dataset statistics in the next version of the paper.
> 3. There was indeed an oversight on line 288. In reality, each essay contains an average of 7 topic sentences. Each secondary school essay consists of 7-8 paragraphs, with about 5 to 8 sentences each. Our annotation aim was to identify the topic sentence for each paragraph. We will add those necessary details to next version.
> 4. The model we utilized was gpt-3.5-turbo. We will rectify this in the subsequent version.
> 5. In lines 468-474, our intention was to highlight that, despite the appearance of randomness in peak performances, essays with higher coherence generally perform better in discourse relation recognition.
> 6. We assure the authenticity of all experimental data. We've also observed the counterintuitive phenomenon you mentioned and discussed it in detail in the appendices E.
> 7. The IAA formula you referenced comes from the paper "What Substitutes Tell Us – Analysis of an “All-Words” Lexical Substitution Corpus" (https://aclanthology.org/E14-1057.pdf). We will incorporate this citation in our next version.
> 8. Thank you for your suggestion regarding the appendices. We plan to integrate its content into the main text.

---

### Meta-Review · Area_Chair_8jtV · 2023-09-18

**Recommendation:** 3

**Metareview:**

The paper reports on a new resource of Chinese middle school essays annotated for three tasks: discourse coherence (global coherence score), topic sentence extraction (identification of the main sentence per paragraph), and, discourse relation recognition (logical relations between sentences). The paper also reports on several experiments validating the corpus using diverse approaches.

Unfortunately, resource papers are not particularly "exciting" as reflected in the lukewarm scores by the reviewers. There seems to be a consensus that the paper needs some re-wording, more in-depth description, and potentially better argumentation, all of which is feasible within the time frame for the final version. In addition, two of the reviewers took issue with what is reported about the corpus construction. Details concerning basic descriptive statistics of the resource are missing and information about how the data was obtained or preprocessed is also missing, e.g, were the essays anonymized? (this last concern is not coming from the reviewers, but from me after perusing the paper). Last, reviewer2's suggestion about highlighting the correlation between the tasks--although somewhat severely worded--would indeed make for a stronger paper. The authors do point to some correlations but structure their discussion per task which reads a bit fragmented.

All in all, I believe this is a valuable resource for the community and that most of the issues can be fixed by rewriting (as opposed to running experiments), hence my recommendation.

---

### Decision · Program_Chairs · 2023-10-07

**Decision:**

Accept-Main

**Comment:**

The paper reports on a new resource of Chinese middle school essays annotated for three tasks: discourse coherence (global coherence score), topic sentence extraction (identification of the main sentence per paragraph), and, discourse relation recognition (logical relations between sentences). The paper also reports on several experiments validating the corpus using diverse approaches.

Unfortunately, resource papers are not particularly "exciting" as reflected in the lukewarm scores by the reviewers. There seems to be a consensus that the paper needs some re-wording, more in-depth description, and potentially better argumentation, all of which is feasible within the time frame for the final version. In addition, two of the reviewers took issue with what is reported about the corpus construction. Details concerning basic descriptive statistics of the resource are missing and information about how the data was obtained or preprocessed is also missing, e.g, were the essays anonymized? (this last concern is not coming from the reviewers, but from me after perusing the paper). Last, reviewer2's suggestion about highlighting the correlation between the tasks--although somewhat severely worded--would indeed make for a stronger paper. The authors do point to some correlations but structure their discussion per task which reads a bit fragmented.

All in all, I believe this is a valuable resource for the community and that most of the issues can be fixed by rewriting (as opposed to running experiments), hence my recommendation.